# Effects of Water and Chemical Solutions Ageing on the Physical, Mechanical, Thermal and Flammability Properties of Natural Fibre-Reinforced Thermoplastic Composites

**DOI:** 10.3390/molecules26154581

**Published:** 2021-07-29

**Authors:** Baljinder K. Kandola, S. Ilker Mistik, Wiwat Pornwannachai, A. Richard Horrocks

**Affiliations:** 1Institute for Materials Research and Innovation, University of Bolton, Deane Road, Bolton BL3 5AB, UK; imistik@marmara.edu.tr (S.I.M.); wiwatpor@scg.com (W.P.); A.R.Horrocks@bolton.ac.uk (A.R.H.); 2Department of Textile Engineering, Faculty of Technology, Marmara University, 34722 Istanbul, Turkey; 3SCG Chemicals Co., Ltd., 1 Siam Cement Road, Bang Sue, Bangkok 10800, Thailand

**Keywords:** natural fibre-reinforced composites, polypropylene, polylactic acid, water absorption-desorption, chemical solution ageing, flammability, flexural properties

## Abstract

Biocomposites comprising a combination of natural fibres and bio-based polymers are good alternatives to those produced from synthetic components in terms of sustainability and environmental issues. However, it is well known that water or aqueous chemical solutions affect natural polymers/fibres more than the respective synthetic components. In this study the effects of water, salt water, acidic and alkali solutions ageing on water uptake, mechanical properties and flammability of natural fibre-reinforced polypropylene (PP) and poly(lactic acid) (PLA) composites were compared. Jute, sisal and wool fibre- reinforced PP and PLA composites were prepared using a novel, patented nonwoven technology followed by the hot press method. The prepared composites were aged in water and chemical solutions for up to 3 week periods. Water absorption, flexural properties and the thermal and flammability performances of the composites were investigated before and after ageing each process. The effect of post-ageing drying on the retention of mechanical and flammability properties has also been studied. A linear relationship between irreversible flexural modulus reduction and water adsorption/desorption was observed. The aqueous chemical solutions caused further but minor effects in terms of moisture sorption and flexural modulus changes. PLA composites were affected more than the respective PP composites, because of their hydrolytic sensitivity. From thermal analytical results, these changes in PP composites could be attributed to ageing effects on fibres, whereas in PLA composite changes related to both those of fibres present and of the polymer. Ageing however, had no adverse effect on the flammability of the composites.

## 1. Introduction

Conventional synthetic fibre-reinforced composites have been a popular choice for transport and construction industries for the last five decades as a consequence of their high strength-to-weight ratios. However, recent pressures towards a circular economy based on increased sustainability and recyclability of the products, etc., has focused interest on the use of natural fibre reinforcements, as well as biopolymer matrices [1,2]. The usage of such natural fibres has gained momentum due to their advantages in terms of low densities, highly specific strengths and moduli, and biodegradability, compared to glass, for example [3,4,5]. Natural fibres may also be used as a secondary reinforcement as hybrids with other synthetic fibres [1,2]. The mechanical properties of composites reinforced with lignocellulosic fibres (such as jute, sisal, flax and hemp), while not sufficiently high for structural applications, are high enough to replace glass fibre-composites for low-cost, low strength applications [6,7,8,9,10,11,12,13].

The main concerns about the usage of these fibres in composites, however, are their poor compatibility with both synthetic and bio-polymers, leading to weak fibre-matrix interfacial adhesion, susceptibility to thermal degradation during melt processing, hydrophilicity leading to moisture absorption, environmental degradation and flammability. To address the weak interfacial bonding, surface modification of fibres [6,14], functionalisation of the polymeric matrix [15] or novel processing techniques such as melt pressing fabrics from co-mingled natural and polymeric fibres [16,17] may be used. The thermal sensitivity of natural fibres at temperatures above 180–200 °C [11] limits the choice of matrix material with melting points being ideally less than 180 °C, which for synthetic polymers focuses interest on polyethylene, polypropylene, polystyrene and polyvinyl chloride, although polypropylene is considered to be the most popular choice [11].

It is reasonable to assume that moisture absorption by natural fibre composites when used in humid conditions could have a detrimental effect on their mechanical properties, and hence their long-term performance. Water uptake by the hydrophilic fibres weakens interfacial bonding with hydrophobic polymers, resulting in reductions in composite mechanical properties [18]. Moisture diffusion can also occur through the micro gaps or voids between the polymer-fibre interfaces [19]. Fibre swelling following moisture sorption can cause further weakening of the interface, leading to cracking of the matrix and delamination [20]. Moisture diffusion in composites depends principally on fibre volume fraction and void volume fraction, humidity and temperature. Composite manufacturing processes may also affect the void volume fraction and hence, the moisture uptake [21]. Several researchers have studied the water uptake of natural fibre composites by immersing in water and studying its effect on composite mechanical properties [5,18,19,20]. If the matrix is composed of a hydrophobic polymer, water absorption within its matrix is negligible, and so is mainly a fibre-dominated property, both within the fibre matrix itself and at the fibre-polymer interface [2,22].

Since the usage of composites is expanding in industries such as marine, oil and gas, chemical plants, offshore industries, etc., where they could be exposed to wet, aggressive environments, their durability to those environmental conditions are extremely important. Composite materials in general are expected to have service lives of about 10–15 years in cars and 25 years in the construction industry. During this period, they are expected to maintain their properties in spite of exposure to a variety of environmental conditions. These include exposures to UV radiation, seawater, acidic and alkaline chemicals, hydrothermal conditions, etc., and combinations of two or more of these. UV radiation exposure most likely affects the polymeric matrix, which is the more exposed part of the composite, thereby sensitising degradation of this phase. However chemical exposure may affect both the fibre and the polymer components. Absorption of chemical solutions occurs similar to water diffusion apart from the presence of additional solute molecules or ions and their potential interactivity. For marine applications ageing in seawater, usually as NaCl solution is undertaken. While some studies have taken place on the effects of sea water and acidic and alkaline environments by immersing natural fibre-composites in those solutions [1,20,23,24], there is still a gap in the understanding of their effects on both physical and chemical changes in individual components, which are determining factors for the mechanical and thermal properties of the composites.

The nature of the polymer matrix and the ageing conditions will obviously affect the ageing behaviour of the composite. Polypropylene is known to be resistant to water, alkaline and acidic environments [25], hence its popularity in natural fibre-reinforced composites [11]. Poly(lactic acid), or PLA, on the other hand may undergo hydrolysis as a consequence of the catalysing effect of free carboxyl end groups produced at elevated temperatures [26] or by changing the pH of the water, which results in polymer chain ester group hydrolysis and associated reductions in molecular weight coupled with other structural changes such as recrystallization [20].

With the aim of investigating the water/aqueous chemical solution ageing effects on composites comprising hydrophilic fiber reinforcing elements and a hydrophobic matrix, a number of natural fibre-reinforced composites have been prepared. These comprised woven jute, sisal and wool fabrics (structures shown in Figure 1) as reinforcing elements within polypropylene (PP) and polylactic acid (PLA) thermoplastic matrices, prepared using a previously reported methodology during which work detailed analysis of the effects of different matrices on the mechanical and fire performances of the composites was reported [17]. The composite preparation method involved needle-punching reinforcing fabrics with polypropylene or polylactic acid fibre webs prior to composite fabrication process using the hot-press technique [17]. The composition of different composites is given in Table 1. All composites were subjected to water, common salt (pH7), acidic (pH4) and alkaline (pH10) solutions for different time intervals up to a maximum of 3 weeks, and the resulting water/aqueous solution uptake behaviours studied. Aged composites were oven-dried to study water desorption as well as possible changes in mechanical and flammability performance. While wool is not commonly used as a reinforcing material due to its lower mechanical properties [26,27], it was chosen with PP for its inherently flame-retardant properties. Wool/PLA composites were not prepared, since wool has a high moisture regain value, which could cause hydrolysis of PLA during melt processing.

## 2. Results 

Visual inspection of the samples after immersion in water/chemical solution (common salt, acidic and alkaline) for 21 days indicated that all jute and sisal composites showed some loss of coloration and milky appearances on the surfaces. Digital and optical micrographs of all composites in Table 1 are shown in Figure 2 before exposure and after exposure to selected solutions for 21 days.

In the digital images (marked (i, ii) in Figure 2), colour changes can be clearly seen in exposed samples, which might be due to the change in refractive index of the swollen fibres after water absorption or to the light scattering of surface effects or internal voids produced at the fibre-matrix interfaces. While the sisal/PP and sisal/PLA composites in Figure 2c,d reflect the open structure of the sisal fabric in Figure 1b, the jute-containing composites (Figure 2a,b) because of the small fibres protruding from the yarn surfaces act as support for respective molten PP and PLA and so respective composites have a web of fibre-reinforced polymer in the coated woven yarn interstices. In exposed samples, the definition of the underlying fibre/yarn structures was lost, particularly in the jute composites, which show a significant colour loss. In sisal composites, due to the open structure of the woven yarns, this effect was not so pronounced. The wool/PP composite, because of the tighter weave of the fabric, was more evenly coated with polymer, and the composite also shows a reduced fabric structural definition after ageing. It should be noted that the reduced clarity in Figure 2e is a consequence of the very similar refractive indices of matrix (*n* = 1.49 for PP, 1.47 for PLA [25,28]) and fibre (*n* = 1.54 for wool [29]), unlike in Figure 2a–d, where the lignocellulose fibres have typical refractive indices ≥1.57 [29]. 

### 2.1. Water/Chemical Solution Sorption-Desorption Characteristics

#### 2.1.1. Relative Sorption Effects

To study water/chemical solution uptake as a function of time, 10 mm × 10 mm composite specimens were immersed in water/chemical solutions for 1, 4, 7, 14 and 21 days at room temperature. The specimens were weighed before and after immersing in water/solution. The apparent weight gain (*W_G_*) values as a function of the square root of immersion time (h) for jute/PP, jute/PLA, sisal/PP and wool/PP composites under all four conditions are shown in Figure 3. It can be seen that in all jute and sisal composites water sorption follows Fickian behaviour, i.e., *W_G_* increases monotonically before reaching a maximum and then equilibrium in PP composites and a subsequent decrease in PLA composites. This is in agreement with the reported behaviour of lignocellulosic composites in the literature [18,30]. The first parts of the curves with rapid *W_G_* increase show the influence of respective hydrophilcities of lignocellulosic and wool fibres. The hydrophilicity of a fibre is due to the presence of the hydroxyl groups in the molecular chain of cellulose in the former and of additional hydrogen-bonding group such as NH and NH_2_ groups in the latter and these groups, coupled with the internal voidage in their respective microphysical structures determines the sorption capacity of water. The quantity of adsorbed water under standard atmospheric conditions (65% rh and 20 °C) is the moisture regain value and the water present is primarily monomolecular water H-bonded to individual functional groups in respective polymer chains. As the humidity increases, the fibre becomes more damp with adsorbed water aggregating into liquid water (often termed physical water), representing the absorption phenomenon. As the relative humidity approaches 100% and fibre surfaces become wet, capilliary water may be absorbed if the swollen fibre structure allows. In lignocellulose fibres like jute and sisal, each fibre is a bundle of microfibers with significant voidage between them. On the other hand, wool, while having a more homogeneous physical structure at the microscale, has an internal morphology at the sub-micro level as well as surface scales [31]. The moisture regain values of jute, sisal and wool fibres typically reported in the literature are 12%, 11% and 15%, respectively for jute, sisal and wool [11,31,32]. However, as seen in Figure 3, the maxima for all curves do not reach these values, because values are not weighted to respective fibre mass percentage contents listed in Table 1. However, the sisal/PP composite in salt water appears to absorb about 12% water, which, if weighted with respect to fibre content, would be closer to 18%. This may be explained by the fact that sisal microfibres are actually hollow [33], as also seen from Figure 4d and so capable of absorbing capilliary water to a higher degree than jute, whose microfibers are not hollow. Furthermore, when in composite form the fibres are encapsulated in the polymeric matrix and water sorption due to capillary action at the fibre surface/matrix interface depends on the fibre surface versus internal structural differences.

Polypropylene and PLA have been reported to have very low (<0.3%) maximum water uptake by various researchers [18,20], indicating that all of the absorption within the composites is through the reinforcing fibres. The Fickian behaviour also indicates that all sorption is by these same fibres. Water absorption by the matrix usually results in micro-cracks at the surface and follows non-Fickian behaviour, as reported by Zhou and Lucas for high temperature water aged carbon fibre epoxy composites [34]. As can be seen in the SEM images of the fractured composites in Figure 4a,g, there are no cracks in the PP matrix, indicating no water absorption. In the PLA composites (Figure 4c,e), while there are no cracks, the matrix surface is very rough, indicating that the matrix has been damaged to some extent.

Further analysis of the water/chemical solution absorption characteristics, of all composite samples in various solutions, was performed by measuring the diffusion coefficient (D) using the method discussed by Zamri et al. [19]. The diffusion coefficient (D), for the composite specimen was calculated using Equation (1):(1)$$D=\pi \;{\left(\;\frac{kh}{4W_{G\left(Max\right)}}\right)}^{2}$$
where *W_G(Max_*_)_ is the maximum weight gain (%), *h* the thickness of the composites and *k* the initial (early stages of water uptake) slope from a plot of %*W_G_* vs. T^1/2^. All of the values are presented in Table 2. 

Jute/PP and Jute/PLA show maximum *W_G_*_(*Max*)_ values for water as 6.3% and 6.0% respectively, which, as reported above, if expressed with respect to percentage fibre content, are in the range of 9.5% and 9%, only slightly lower than the moisture regain value of jute (12% [11,32]), which of course will be higher at 100% relative humidity or when wet. However, these values are in a similar range to those reported by others [5] for jute composites and any differences would depend on the fibre content and resin matrix interface, as well as the composite preparation method. The absorption behaviour of each chemical solution is very similar to that for water for all composites in Figure 3. This same effect can also be seen in Figure 5a where maximum water/chemical solution uptake values, *W_G_*, for all larger samples (having the required dimensions for flexural and flammability testing) are presented. Any small differences in the maximum values of water absorption for small samples in Table 2 and large equivalents in Figure 5 are due to the differences in geometry of samples. In Figure 5b, the mass changes after drying in an oven at 50 °C for 24 h, *W_L_*_,_ after which all the adsorbed water should have desorbed are plotted, which show overall small mass losses (<1 wt%) for the variously exposed jute-reinforced composites. 

Fickian behaviour was also seen in the case of the sisal composites, but after reaching a maximum, *W_G_* decreased with further ageing; similar behaviour was also seen by Chow et al. [18]. The water and chemical solution uptakes for sisal/PP and sisal/PLA in Figure 3c,d are very different, and the results for sisal/PLA are particularly erratic, when compared to the results for larger samples in Figure 5a. The smaller samples are very thin and the yarns, or indeed fibres, appear as monofilaments (see Figure 1b), as reflected in the composite micrographs in Figure 2b. Sisal fibres, being hollow, are subject to capillary action, which maybe significantly influenced by the ionic characteristics of the solutions. Smaller samples under the swelling forces of water sorption, are more likely to promote significant sample deformations than in larger samples, which could affect their sorption properties. That the larger scale results in Figure 5a appear to be more self-consistent, supports this suggestion. 

The water uptake is much higher in sisal composites than jute composites (see Figure 5a), which is most likely a consequence of their hollow microfiber character and despite their reported moisture regain values being very similar (11% and 12% respectively [11,32]). However, it must be remembered that both lignocellulose fibres contain significant levels of hydrophobic lignin and potentially water soluble hemicelluloses, which will influence water sorption characteristics. Similar high water uptake by sisal composites compared to jute fibre composites has also been reported by Carvalho et al. [5].

In case of wool/PP samples, while the *W_G_* increased monotonically, equilibrium was not reached (see Figure 3d), indicating non-Fickian behaviour and maximum water sorbed, *W_G(Max)_*, is ~6.5%. However, while wool had the highest moisture regain (15% [31]) of the fibres used, it might be expected that the wool/PP composite would adsorb more water than sisal/PP and jute/PP composites. However, Table 2 and Figure 5a show that the order of increasing water content is sisal/PP > jute/PP > wool PP. The highest value for sisal/PP is obviously related to the hollow microfiber structure, but the lower behaviour of wool/PP relative to jute/PP composites may be linked to the presence of relatively hydrophobic external scales present on wool fibres, which reduce water absorption at the matrix/fibre interface.

#### 2.1.2. Effect of Chemical Environment

Over the pH range used to expose all composites (pH range 4–12) at room temperature, it may be assumed that either polymer matrix, PP or PLA, will not be affected, and so any chemical interactive effects will be solely related to the relative fibre-chemical reactivities. Cellulose is generally resistant to alkaline conditions under anaerobic conditions, but may be acid-catalysed at pH4 at elevated temperatures [35]. While lignin present in both jute and sisal will also be generally inert under these conditions [36], the presence of soluble, low molecular weight hemicelluloses may be influenced by pH via formation of more soluble sodium salts at pH > 7. Wool on the other hand is generally considered to be acid resistant in terms of hydrolysis at pH 4–7, but when pH > 7 cystine disulphide cross-links may be hydrolysed, which cause both physical swelling and partial dissolution [35]. Wool also contains free acidic (e.g., –COOH) and basic (e.g., –NH_2_, –NH–) groups, which self-neutralise at the so-called ioselectric at pH 4.2–5.0 [37].

Against this background, the increases in water sorption in Figure 3a,b and diffusion coefficients in Table 2 for small sample jute/PP and sisal/PP composites respectively, in the presence of all three solutions, may be considered. Surprisingly, however, the presence of salt produced the greatest effect, which suggests that normal sorption at pH7 is significantly increased within a neutral ionic environment. Compared with jute/PP, sisal/PP composite sorption is much less influenced by either acidic or alkaline environments within experimental error and this may be due to the over-riding effect of capillary absorption within the fibre hollow morphology masking any lesser effects of pH. However, the diffusion coefficients in Table 2 show that sisal/PP values increase under both salt and acidic conditions. Figure 5a however, for 21-day water sorption data for larger samples, shows little effect of the chemical environment for both jute/PP and sisal/PP except for the reduced sorption under alkaline conditions for the latter.

Jute/PLA behaviour for composite samples of both sizes suggests that both salt and acidic environments reduce composite sorption behaviour, an opposite effect to the jute/PP composite and one that is difficult to explain. PLA, however, is sensitive to acidic hydrolysis, and so some of the matrix might have been solubilised, thereby reducing the observed overall weight gain from water sorption alone. Diffusion coefficients, however, show that presence of salt and acid reduce values, while alkali shows no change in agreement with the 21-day sorption data in Table 2.

Figure 3e for wool/PP suggests that, within the error margin, an alkaline environment surprisingly does not influence water sorption properties, with both salt and acid solutions showing slightly reduced sorption at longer times. Figure 5a for the larger samples, shows both alkaline and salt conditions slightly increasing 21-day sorption levels. These latter results reflect the diffusion coefficient in Table 2, which suggest that both salt and alkaline conditions more than double the respective solution diffusion coefficients for wool/PP with respect to water alone, as might be expected from the alkaline sensitivity of wool and its zwitter ionic character. 

Clearly the rates of water diffusion and maximum sorption data under different pH conditions are not easy to explain and besides being determined by fibre chemical characteristics in the main, will be influenced by respective fibre physical morphologies, as will be observed in Section 2.2.

#### 2.1.3. Relative Desorption Effects

It can be seen in Figure 5b that, on drying the aged composite samples in oven at 50 °C for 24 h, all of the absorbed water, including water from chemical solution-exposed samples, was desorbed in jute or sisal fibre-reinforced composites, but not entirely from wool/PP composites, possibly because drying time was insufficient for full moisture loss and the greater regain of wool fibres gave rise to greater hysteresis effects and so the development of internal voidage. While the increase in sample mass after ageing and drying observed in Figure 5b may be ascribed to the sorption-desorption hysteresis of wool, these increases are little influenced by pH, confirming that water alone is the major cause.

However, the mass losses after ageing treatments noted for all composites containing jute and sisal fibres suggest that there has been removal of some of the respective fibre hemicellulose fractions, which are small oligomers with varying degrees of water solubility. In both jute and sisal, values are about 14% by weight of the total fibre [11,32]. Table 1 shows that the mass fractions of jute and hemp are about 40%, and so the small mass losses recorded in Figure 5b could easily be explained in terms of hemicellulose loss. However, Figure 5b does show that after drying sisal-containing composites generally lose greater weight than jute-containing ones, a possible factor again being their hollow character, which increases the possibility of hemicellulose removal. Since the mass losses are 1% or less, within error, it is difficult to note whether the slight differences with solute type are significant.

### 2.2. Effect of Ageing on Mechanical Properties of Composites

To establish the potential usefulness of each composite in terms of a semi load-bearing structural component, the flexural properties of unaged and water/chemical solution-aged samples were measured in a three point bending mode. The flexural moduli of samples tested in the elastic region only were calculated using the Engineers’ bending theory [38], the results are given in Table 3.

The flexural properties of all control samples, except for the wool/PP composite, have been discussed in detail elsewhere [17], where it has been explained that jute or sisal/PP composites (2.6 and 3.2 GPa) have lower flexural moduli than the respective PLA composites (6.0 and 5.9 GPa). This is in accordance with other findings in the literature, which show higher mechanical properties of PLA than PP [39]. Amongst jute-, sisal- and wool-reinforced PP composites, the flexural moduli of jute/PP and wool/PP are similar (2.6 and 2.2 GPa), with sisal being higher (3.3 GPa). The differences are due principally to different sample thicknesses, and fibre volume fractions (see Table 1) and other factors, such as reinforcing fibre modulus.

After ageing in water/chemical solutions flexural modulus of each sample generally decreased compared to the respective control. While, the absolute values of the latter are given in Table 3, percentage reductions in flexural modulus with respect to the respective control sample for all air dried and oven dried samples are given in Figure 6a,b respectively. As can be seen from these results, the percentage reductions in air dried or oven-dried samples are similar. This shows that even when the water is desorbed after oven-drying (Figure 5b), the mechanical properties of the original composites are not restored.

On comparing the effects of different ageing solutions in PP composites, we found that the effects of each solution type are similar to that of water alone, considering the variation within results. However, in wool composites there is greater reduction in flexural modulus in salt, acidic and alkaline media, in the decreasing order of effect as: acidic > salt ~ alkali > water. Comparing each fibre reinforcement type in PP composites, it can be seen that modulus reductions in sisal/PP composites are much greater than in jute/PP, whereas they are much lower in wool/PP composites. 

Generally these reductions may be related to the water sorption properties of these fibres within the composites as shown in Figure 3 and Figure 5a,b. Under wet conditions, saturation of hydrogen-bonding sites coupled with capillary action into fiber morphological voids will promote swelling and increasing internal stresses within the constraining matrix polymer. On the one hand, both adsorbed and absorbed water may have a plasticizing effect within the fibre morphology [31], as well as affecting the interfacial cohesion between the fibre surfaces and the matrix on the other [18]. Assuming that such swelling will be accompanied by irreversible changes in fibre and fibre-matrix morphologies, after desorption such changes will be reflected in dried composite mechanical properties. Thus, when the water is desorbed, the composite cannot simply return reversibly to its original physical form, hence explaining the non-restoration of flexural moduli recorded in Figure 6. This degree of irreversible structural change may thus be related to the percentage flexural modulus reduction for each composite/ageing medium combination, as can be seen in Figure 4, where considerable voidage at the fibre-matrix regions can be observed. Such increases in voidage would be expected to show as a loss of transparency or increase in light scattering of aged, dried samples and inspection of selected aged samples in Figure 2a–d shows a reduction in clarity of the underlying yarn/fibre structures with respect to the control sample optical micrographs.

While it is generally evident from the results in Figure 5a and Figure 6 that percentage water absorption, *W_G_*, values appear to be related to reductions in flexural modulus, the addition of solutes only has a minor additional effect. Figure 7 presents plots of the 21-day water absorption values in Figure 5a versus percentages reduction in flexural modulus, ΔFlex, between the control composites and those having been exposed to the different solutions.

In Figure 7a, for the control composites, a clear W_G_ versus ΔFlex relationship independent of the composite type can be observed. The effects of salt, acidic and alkaline solution exposures in Figure 7b–d respectively show the same linear relationship. When superimposed in Figure 7e, a general trend remains with the effects of each solute having very slightly different slopes. These trends suggest that the changes to the composite structure as a consequence of water sorption and the consequential irreversible effect that it has on the fibre and fibre-matrix interface morphologies are the cause of the reductions observed in flexural properties.

The degree of fibre-PP matrix cohesion within each composite will depend on formation of strong intermolecular forces across the interface. At first glance it might be assumed that the cellulosic presence in sisal and jute and the keratin protein content in wool will yield similar levels of poor cohesion because of the hydrogen-bonding potential of each fibre type, contrasted with the van der Waal’s force potential only in PP. In considering this interface for wool/PP, however, scales on the wool fibre surfaces are extremely hydrophobic, which will enhance fibre-PP cohesion via van der Waal’s force attractions compared to the PP composites containing cellulosic fibres. The Wool/PP composite also shows lower overall flexural modulus hysteresis (see Figure 6), suggesting that moisture adsorption/desorption is a more reversible process than observed in the other composites. The less efficient fibre-surface cohesion in the sisal/PP and jute/PP composites coupled with their respectively much higher flexural property hysteresis, suggest that it is this interface and the absorption of water within it coupled with high levels of potential swelling discussed in Section 2.1.1 that will be dominant features. The fact that the wool/PP composite has the lowest diffusion coefficient (Table 2) also suggests that the composite is adsorbing water principally via fibre and not via the fibre-PP interface diffusion. For the sisal/and jute/PP composites the effects of lignin content present in both jute (13%) and sisal (11%) [40] may also influence fibre/PP cohesion. The presence of this hydrophobic polymer will enhance fibre-PP cohesion to a greater extent in the former with respect to the latter and may explain why the jute/PP composite has similar water uptake to the wool/PP composite.

With regard to the added effects of solute present in each ageing solution, the results in Figure 3 and Table 2 show that both solute presence and accompanying pH, while modifying the water-absorbing properties of the fibre, as discussed in Section 2.1.2, also add further to flexural modulus hysteresis, as seen in Table 3 and Figure 6 and Figure 7, probably by increasing internal stresses within the composite, although the effects are much less extensive than those caused by water alone. Within error, while all composites except sisal/PP show little or no significant increase in either water/solution absorption in the presence of salt (Table 2). Figure 6a suggests that while further reductions in flexural modulus are negligible in sisal/PP and jute/PP composites, it is significantly reduced in the wool/PP composite after all three solution exposures. Salt treatment of cellulosic fibres can remove some hemicellulose, lignin, wax and pectin [41], although at room temperature this effect is expected to be very low. As discussed in Section 2.1.2, wool with an isoelectric pH range 4.2–5 [37] would be expected to absorb sodium and chloride ions at pH7, and it has been reported that sodium ions can promote the degradation of disulfide groups in the keratinous protein of the wool [42], resulting in reductions in fibre strength. The addition of acid slightly further reduces the flexural modulus of jute/PP and sisal/PP but significantly so for wool/PP composites, again emphasizing the ionic character of the latter fibre content, with possible breaking of internal ionic cross-links in the fibre structure. As stated in Section 2.1.2, while cellulose is well known to be hydrolysed under acidic conditions, especially when pH < 4, under ambient temperature, pH4 conditions, such hydrolysis may be considered to be negligible since acidic exposure has only a marginal effect on flexural rigidity of the respective derived composites. The effect of alkali at pH10 has little or no effect on sisal/PP and jute/PP flexural behavior, but significantly reduces that of the wool/PP, as might be expected from its alkaline sensitivity in the composite, although again this effect is removed following oven drying. 

There is also the possibility that some hydrolytic dissolution of the PLA matrix may occur on immersion in water, but this occurs usually only at elevated temperature or after very prolonged exposure, resulting in reduction of the molar mass [20,25]. The adsorbed water can also have a plasticizing effect on the PLA matrix, but this is usually a reversible effect and on drying should be removed [20,43]. Deroine et al. and many other researchers have studied the ageing of PLA in water for prolonged periods, ranging from 1 to 6 months [44] and references [14,15,16,17,18,19,20] therein. While a reversible degradation at short times and at lower temperature was observed, the loss of properties was quite small even after 6 months’ exposure at 25 °C. However, there are also some reports where reductions in molecular weight, even at 20 °C, have been reported [20]. As can be seen from Figure 6a,b and Figure 7 the reduction in flexural modulus of jute/PLA is related to the water uptake, sisal/PLA sample shows anomalous behavior for small samples in Figure 3e and is an outlier for larger sample exposure in Figure 7a. This could be due to the open structure of the sisal reinforcement (Figure 1b and Figure 2c,d), leaving PLA matrix more exposed and PLA hydrolysis could explain this observed erratic behavior.

Mineral salts can facilitate the diffusion of water in the polymer, leading to hydrolysis [43]. Acid and alkaline solutions also catalyse hydrolysis and depolymerization of PLA [45]. As can be seen from Figure 4c,e, the PLA matrix surfaces are much rougher with microcracks appearing after water and solution immersion compared to rather smooth surfaces in respective control sample, supporting the above arguments. Typical of other polyesters such as poly(ethylene terephthalate), the catalyzing effect of alkali is more than that of an acid at similar concentrations [45]. In the PLA composites studied here, while the effects of sodium chloride and acidic solutions appear to be minimal in terms of further reduction in respective flexural moduli relative to the respective sisal/PLA and jute/PLA controls, there is more reduction in the alkaline medium. This perhaps reflects the alkaline catalyzed hydrolytic sensitivity of PLA mentioned above.

### 2.3. Effect of Ageing on Flammability Properties of Composites

#### 2.3.1. Limiting Oxygen Index (LOI)

The LOI values of unaged and aged composites are presented in Table 3. Both Jute/PP and sisal/PP presented LOI = 18.7%, which indicates their relatively high flammability. Wool/PLA has a value of 20.7, which suggests slightly lower flammability than jute or sisal/PP composites, most likely a consequence of the inherent flame retardance of the wool fibres which typically yield LOI values ~25%. All PLA composites have slightly higher LOI values than respective PP composites, which is mainly due to lower flammability of PLA (LOI = 20%) than PP (LOI = 17.4%), been discussed in detail elsewhere [16,46].

As can be seen in Table 3, water absorption, followed by desorption, has not adversely affected composite flammability, with on average all aged samples increasing values by 0.1 LOI unit with respect to respective controls. This difference is, though very insignificant, significant in that it demonstrates an increase rather than a decrease. The irreversibility of the water sorption-desorption experience, which will be accompanied by slight composite structural changes with a possibly more open structure, will be an influencing factor here. Salt solution exposure of all jute or sisal composites has increased the LOI by about 0.4% units with respect to water-only exposed samples, which is possibly due to the residual Cl^−^ ions present, which may react with the decomposing fibre and act in gas phase, reducing flammability. Acidic solution exposure has no further effect on LOI of all jute and sisal composites. All Lewis acids catalyse dehydration -based decomposition of cellulosic fibres as opposed to depolymerisation reactions, leading to more char formation and reduced flammability [46]. Hence, acidic solution exposure is not expected to adversely affect the flammability of these composites. Alkaline solution exposure has also minimal further effect on all PP and PLA composites. In alkaline medium some hemicellulose, lignin, pectin, etc., could dissolve, which would be expected to slightly increase the flammability, although no such effect is seen here.

For the wool/PP composite, exposure to any of the chemical solutions has a minimal further effect on LOI with respect to water-only exposure. This is perhaps surprising, given the higher regain of wool, its sorption-desorption hysteresis and the consequent retention of water after drying, as noted in Figure 5b.

#### 2.3.2. Flame Spread

The UL-94 vertical burn test results for jute- and sisal-reinforced PP and PLA composites have been explained elsewhere [17] and because they were completely burnt, in this work they were only tested in horizontal orientation. The rates of flame spread of the neat polymers and composites were studied by recording times taken to reach 50 (T_1_) and 100 mm (T_2_) marks in both horizontal orientations, and the results are presented in Table 3. As can be seen, that water sorption-desorption reduced the flame spread of jute/PP and jute/PLA composites and exposure to the chemical solutions slightly increased respective values. In sisal-reinforced composites both water and any chemical solution exposure slightly increased the flammability shown as slightly increased burn rates. This is contrary to the LOI results. However, both tests are different and the samples burn differently, hence some differences in behaviour are expected. Overall, it could be concluded that water or chemical solution ageing of all of these composites has no significant effect on their flammability. In all jute or sisal composites there was no melt dripping observed, most likely because the entrapped fibres charred to produce a scaffolding to support any molten polymer.

In the case of wool/PP composites there was observed considerable melt dripping with up to 216 drops able to be counted. In this case water alone and all chemical solution exposures reduced the flame spread rate with respect to the control. Water and acidic solution had slightly more effect than alkaline or salt solutions. Given that wool also tends to char when heated and does so following a semi-liquid state [47], this latter property will not prevent the molten polypropylene from melt dripping. Furthermore, this enhanced melt dripping effect is particularly sensitive to both sodium chloride and alkaline solution exposures, most likely because of their interactions with wool, as discussed above. 

### 2.4. Effect of Ageing on Thermal Properties of the Composites

To understand the effect of ageing on the polymeric matrix or the fibre component, thermal analysis of each neat polymer, fibre type and derived composite was performed in flowing air. To observe whether there is any interaction between the two components of the composite at the processing stage, the calculated weighted average curves from the respective curves of individual components were compared with the experimental curves. From the experimental curves of each composites, specific parameters were then chosen and the effect of ageing on those parameters were studied. TGA and DTA curves of fibres, polymers and composites are given in Figure 8 and the derived data presented in Table 4. 

Lignin is reported to undergo thermal degradation from about 250 °C onwards [48] and so it is likely that its presence in both jute (~11%) and sisal (~13%) [11,40] will influence the thermal response of both fibres, and, indeed, both start losing mass above ~250 °C, which is much higher than the melting point of PP or PLA (Table 4). TGA curves in Figure 8 show that both fibres decompose in three stages, leaving no char residue above 450 °C, which is typical behaviour of lignocellulosic materials [11]. These mass loss stages relate to two or three exothermic peaks in the DTA responses (see Table 4), which represent decomposition and char oxidation reactions for these fibres. For TGA curves, the temperatures at which 10% (T_10_) and 50% (T_50_) mass loss occurs represent respective onset of decomposition and maximum mass loss conditions, and are presented in Table 4. 

Wool, on the other hand, starts losing mass above 210 °C, but the rate of mass loss is much lower than for jute and sisal as a consequence of its complex polypeptide structure containing relatively reactive cystine cross-links and other side groups such as –NH_2_. For example, at about 250 °C, hydrogen sulphide is released, and following further disulphide bond interactions char formation occurs [49]; the presences of sulphur and nitrogen are largely responsible for the relatively low flammability of wool.

The polypropylene TGA curve in Figure 8b shows 98% mass loss in the temperature range 220–395 °C, representing decomposition via random chain scission, followed by 2% mass loss over the range 395–450 °C, representing oxidation of residual combustible products. No char was left at the end of the test. The TGA curve of PLA in Figure 8b shows that the mass loss started at 296 °C and 97% mass loss occurred up to 385 °C, followed by a further 3% up to 465 °C. On comparing TGA and DTA curves of PP and PLA, the PLA can be seen to be slightly more thermally stable than PP.

The thermal analytical behaviour of jute/PP and jute/PLA composites is a combination of their respective components. The detailed analysis is given in Table 4, while exemplar calculated weighted average TGA curves for jute/PP and jute/PLA from the respective curves of individual components are shown in Figure 9a,b. The calculated and experimental curves are very similar, indicating that both components are decomposing individually with little or no chemical interaction between them at this stage. Similar behaviour, as exemplified by Figure 9, was shown by sisal/PP, sisal/PLA and wool/PP combinations. The thermal stabilities of different fibre/PP or fibre/PLA composites were determined by the thermal stabilities of the respective fibre types present in the main, i.e., jute/PP or PLA composites were found to be slightly more thermally stable than the respective sisal composites, and wool/PP composites are the most stable. 

To investigate the effects of ageing, selected thermal parameters from Table 4 were chosen. For TGA curves T_10_ and T_50_, and from DTA a characteristic, sharp decomposition peak (T_DP_) for each composite was chosen, which is highlighted in Table 4. The effects of ageing on these parameters are presented in Figure 10.

As can be seen in Figure 10(a1–a3) for the jute/PP composite, in general T_10_, T_50_ and T_DP_ values increase after water and all chemical solution ageing conditions, though the effect in water is minimal. In all cases the effect is greater during the first 7 days, and then little change occurs for longer periods, consistent with respective water adsorption properties seen in Figure 1. PP is generally considered to be resistant to water, salt, mild alkali or acidic solutions at room temperatures [25], hence the matrix should not be affected and so any observed effect is from the presence of jute fibres. As discussed previously, water sorption will most likely have a plasticizing effect on the composites, and on desorption irreversible structural changes at the fibre/polymer interface causing voiding, as shown by the loss of clarity in dried composite photomicrographs in Figure 2 and SEM images in Figure 4. Salt solution exposure appears to increase T_10_, T_50_ and T_DP_ values slightly, while alkaline and acidic solutions have more significant effects in apparently stabilising the fibre. In general, the effect in decreasing order is alkali > acid > salt > water. It must be noted that during the sorption/desorption/drying cycle, any soluble metallic salt impurities which might otherwise catalyse thermal degradation may also be removed. Furthermore, formation of sodium-cellulose salts will stabilise cellulose [50]. In addition, some lignin, hemicellulose, pectin and wax from lignocellulosic fibres (jute and sisal) can be dissolved in alkaline solution [43] and since all of these components decompose at lower temperature than the cellulosic component, their removal would increase the T_10,_ T_50,_ etc. Salt treatment of cellulosic fibres can also remove some lignin, hemicellulose, etc. [41], though at room temperature, this effect is expected to be very low, and hence a marginal effect is seen here. 

Salt water ageing appears to have elevated T_10,_ T_50_ and DTA 1st peak temperatures during the first 7 days’ exposure for jute/PP composites, but reduced values over the whole 21 days for jute/PLA composites. Sisal/PP composites show only reduced T_10_ values while sisal/PLA shows similar behaviour to jute/PLA composites, which suggests that sodium chloride has interacted with the PLA matrix. In the wool/PP composites, similar reductions in all three parameters are noted in all likelihood to be a consequence of salt-wool interactions. As noted in Section 2.3, residual Cl^−^ ions can help in reducing the combustion properties of both fibre and matrix, and it is most likely that their effect in sensitising decomposition for jute/PLA, sisal/PLA and wool/PP composites will be a part of this flame retarding effect.

In the case of acidic solution ageing, results were somewhat mixed, with stabilisation of jute/PP and sisal/PP composites being observed, possibly as a consequence of removal of impurity metal ions as previously noted. Jute/PLA and sisal/PLA composites appear to have been destabilised following acidic ageing, which could be a consequence of acid attack on the PLA matrix. The wool/PP composite shows only significant increases in T_10_ and T_50_ values after 21 days exposure with variable changes in the DTA 1st peak temperatures, which could be related to the ionic character of wool, which at pH4 will have quaternised otherwise free amine groups present to –NH_4_^+^ species, and so modified its thermal decomposition pathway.

## 3. Discussion and Conclusions

In this work, jute, sisal and wool fibre-reinforced PP and PLA composites were subjected to water, salt water, acidic and alkali solution immersion treatment for periods up to 21 days and the effects of water/chemical solution uptake on mechanical and flammability and thermal behaviours of the composites were studied. In all composites the water uptake reflected the respective fibre hydrophilicities and exceeded normal standard regain values when values were weighted with respect to fibre contents. This would be expected under wet conditions, when capillary water sorption into respective fibre voided morphologies is taken into account. This was especially noted in sisal-containing composites which had hollow fibre morphologies, and so greater capillary-absorbing capacity. 

The evidence presented suggest that the accompanying fibre swelling has generated stresses, which together with the individual fibre sorption-desorption hysteresis effects, have promoted fibre surface-matrix separation, showing itself as void formation, which was observed as an increased opacity of dried composites and directly by SEM. Following desorption and drying at 50 °C for 24 h, all composites except wool/PP showed losses in mass attributed to the removal of water-soluble hemicellulose fractions. Wool/PP composites on the other hand showed small increases in mass as a consequence of sorption-desorption hysteresis. In all cases, the flexural modulus of treated composites was reduced and a linear relationship within flexural modulus reduction and water adsorption/desorption was observed, which indicated that the irreversible structural changes, especially at the fibre-matix interfaces, coupled with sorption-desorption effects significantly reduced mechanical performance. The effect of pH and related solutes present caused further but minor effects in terms of moisture sorption and flexural modulus changes. PLA composites were affected more than the respective PP composites because of their hydrolytic sensitivity. While the effects of water/chemical solution ageing have no adverse effect on composite flammability compared to the unaged samples, thermal analytical behaviour enabled the effect of solutes on different components in each composite to be better understood. In PP composites these changes could be attributed to ageing effects on fibres, whereas in PLA composites changes related to both those of fibres present and of the polymer. Given the important relationship between moisture sorption and its associated swelling stresses promoting void formation at the fibre-polymer interface with consequent loss in mechanical properties, a major conclusion is that the role of the moisture sorbing properties of naturally occurring fibres on ageing properties during service should be of major consideration when they are used to replace inert fibre reinforcements such as glass.

## 4. Materials and Methods

### 4.1. Materials

Loomstate, plain woven jute (174 gm^−2^), sisal (62 gm^−2^) and wool (172 gm^−2^) fabrics were used as reinforcements for the production of textile preforms. Jute and sisal fabrics were sourced from the National Institute of Textile Technology Research and Design (NITTRAD), Bangladesh. 

Polypropylene and biodegradable polylactic acid fibres were used as matrices for the production of thermoplastic composites. The fibre staple length of polypropylene was 50 mm and linear density was 3.3 dtex, and the staple fibre length of biodegradable polylactic acid was 40 mm with linear density of 2.2 dtex.

### 4.2. Production of the Textile Preforms and Composites

The detailed procedure of the production of the composites is presented elsewhere [17] and summarised here. Firstly, textile preforms were produced in a two-step process in which: (i) nonwoven webs were produced from PP and PLA fibres by using an Automatex carding machine; and (ii) each nonwoven web and the reinforcing fabric in 40:60 (*w*/*w*) ratio were fed together to the Automatex needle-punching machine. The machine parameters used are presented elsewhere [17]. 

Composite materials were produced from these preforms by melt pressing techniques. Eight layers of each fabric were placed between two aluminium plates, and heated at 190 °C for 2.5 min under 20 kg/cm^2^ pressure. The compressing plate-composite assembly was transferred for cooling to another press operated under water cooling conditions for 2 min at 10 kg/cm^2^ pressure. 

### 4.3. Ageing Studies

Ageing was undertaken in four different aqueous media, namely water and common salt, acidic and alkaline solutions at room temperature, which reflect the extremes most probably existing within a natural environment, namely a pH range 4–10. The water used was distilled water, the salt solution was prepared from 10% sodium chloride, the acidic solution comprised 10.12 g/L (0.05 M) potassium hydrogen phthalate with resulting pH value of 4 and the alkaline solution comprised 4.2 g/L (0.05 M) sodium hydrogen carbonate and 4.0 g/L (0.1 M) sodium hydroxide to yield pH value of 10. All chemicals were sourced from Fisher Scientific and of technical grade. 

Two sets of samples were aged for uptake and ageing respectively during water/chemical solution exposures. To study water/chemical solution uptake as a function of time, 10 mm × 10 mm composite specimens with the thicknesses defined in Table 1 were immersed in each aqueous medium for 1, 4, 7, 14 and 21 days at room temperature (18–20 °C). Control specimens were dried at 50 °C for 24 h and weighed before immersion. After set times, the immersed samples were removed from each water/chemical solution bath, rinsed in distilled water, air dried and weighed. Water (or chemical solution) uptake was determined by measuring the increase in sample weight as shown previously in Equation (2):(2)$$W_{G}=\frac{W_{t}-W_{0}}{W_{0}}\times 100$$
where *W_t_* is the weight of the sample after *t* days/hours exposure to water or solution and *W*_0_ is the weight of the sample before exposure.

To study the effect of ageing on mechanical and fire performance, larger samples of the dimensions required for flexural and LOI testing, were immersed in water/chemical solutions for up to a maximum of 21 days (504 h). After set periods, specimens were removed, air dried and weighed to measure the water or solution uptake using Equation (1). Samples were then dried at 50 °C for 24 h in a heated oven to remove the sorbed water and weighed again. Desorption of water or chemical solution was measured in terms of weight loss (*W_L_*) as:(3)$$W_{L}=\frac{Wo-Wd}{W_{0}}\times 100$$

### 4.4. Flexural Testing

The flexural moduli of all dried samples before and after ageing in water/chemical solutions for 21 days (air dried and oven dried) were determined in a three-point bending according to BS EN ISO 14,125 in the elastic region using an Instron 4303 universal testing machine. A 100 N load cell with a compression rate of 1 mm/min was used for samples with span lengths of 100 mm. Two replicate specimens, with dimensions of 120 mm × 25 mm × thickness as defined in Table 1 for each sample, were tested and the results were averaged. 

### 4.5. Flammability Analysis

A Fire Testing Technology Limiting Oxygen Index Tester was used to carry out limiting oxygen testing according to ISO 4589 on samples of dimensions 12.5 mm × 100 mm × thicknesses as in Table 1.

Flame spread tests were undertaken using a modified UL-94 test in which specimens were strands (length = 120 mm, width = 10 mm) to observe their burning behaviour in a horizontal orientation. Each specimen was clamped horizontally and subjected to a flame of 20 mm height using a Bunsen burner and keeping a 10 mm distance between the end of the specimen and the top of the Bunsen burner. A thin layer of cotton fibres was positioned 300 mm below the test specimen in order to catch molten/flaming drops. The first 10 mm length of sample burning was not taken into account, and so times of burning were recorded once the flame had reached a line drawn 10 mm from the edge against which a flame of 20 mm was applied for 10 s, as specified in the test. A video recording was taken of the burning behaviour of each sample, from which times to reach 50 (t_1_) and 100 mm (t_2_) marks and/or to achieve flameout were noted. Two replicates of each sample were burned and results averaged. The burning behaviour of each sample was observed and noted according to the standard. 

### 4.6. Thermal Analysis

Differential thermal (DTA) and thermogravimetric (TGA) analyses of the composites were performed using an SDT 2960 Simultaneous DTA-TGA instrument (TA Instruments, Elstree, Hertfordshire, UK) on 10–15 mg sample from room temperature to 700 °C at 10 °C/min heating rate under a flowing air atmosphere (100 mL/min). DTA-TGA properties of unaged, 1 week (168 h) aged and 3 week (504 h) aged samples were investigated. Before DTA-TGA analysis composite samples were dried in an oven at 50 °C for 24 h. 

## Figures and Tables

**Figure 1 molecules-26-04581-f001:**
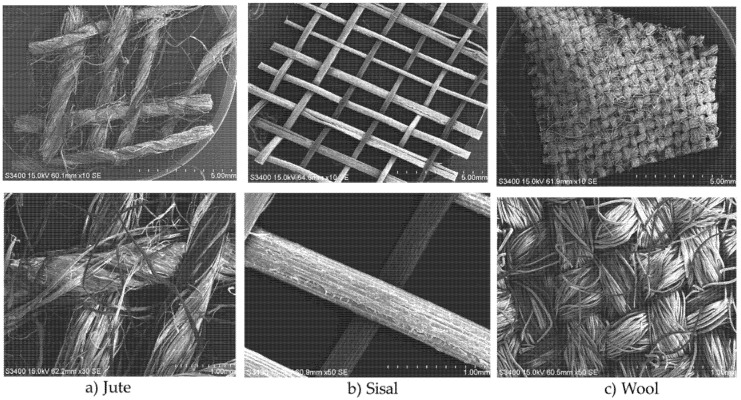
SEM images of (**a**) jute, (**b**) sisal and (**c**) wool woven fabrics.

**Figure 2 molecules-26-04581-f002:**
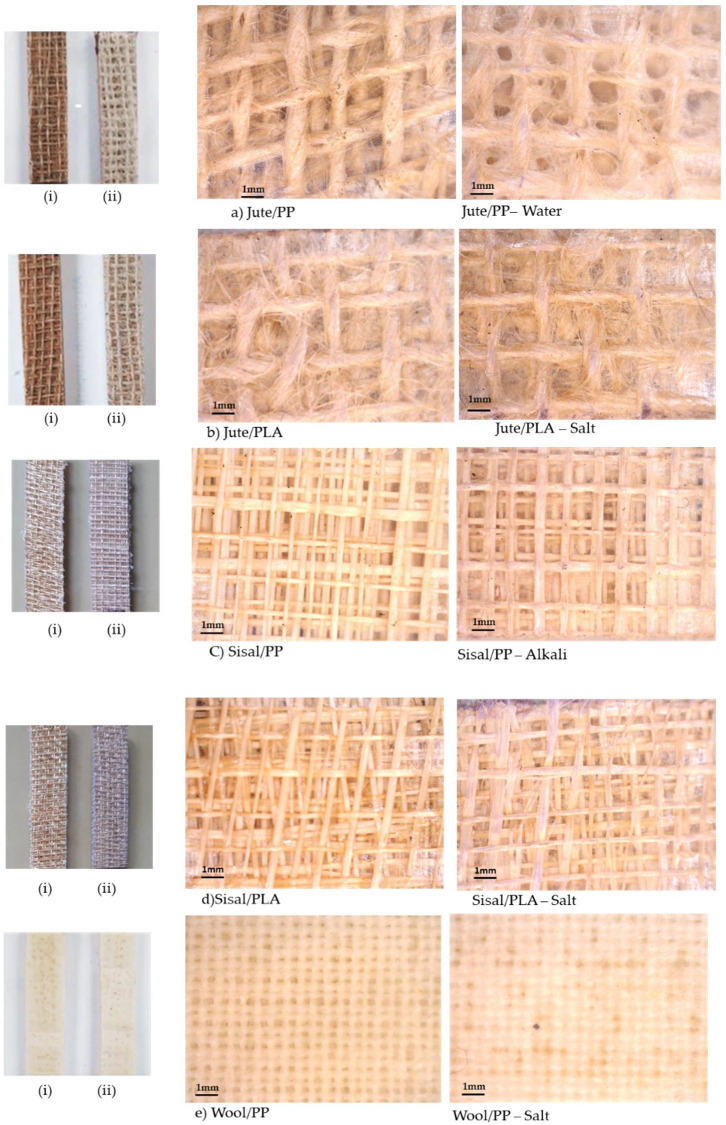
Visual (**i**—before exposure, **ii**—after exposure) and optical micrographs of dried (**a**) jute/PP, (**b**) Jute/PLA, (**c**) sisal/PP, (**d**) sisal/PLA and (**e**) wool/PP composites before exposure and after exposure to exemplar water or solutions for 21 days.

**Figure 3 molecules-26-04581-f003:**
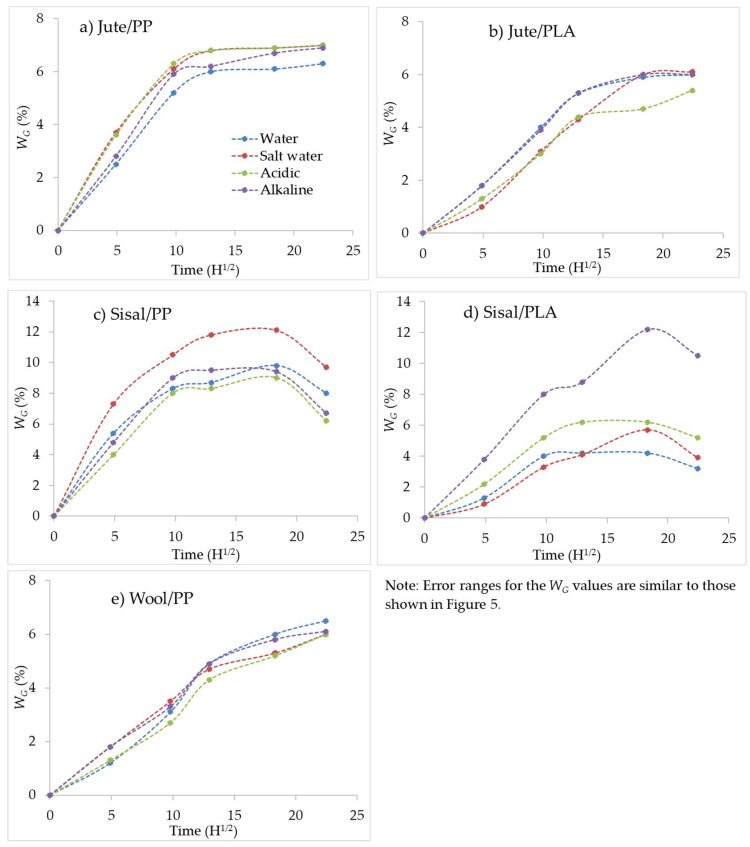
Apparent weight (W_G_) gain versus square root of immersion time for (**a**) jute/PP, (**b**) jute/PLA, (**c**) sisal/PP, (**d**) sisal/PLA and (**e**) wool/PP composites.

**Figure 4 molecules-26-04581-f004:**
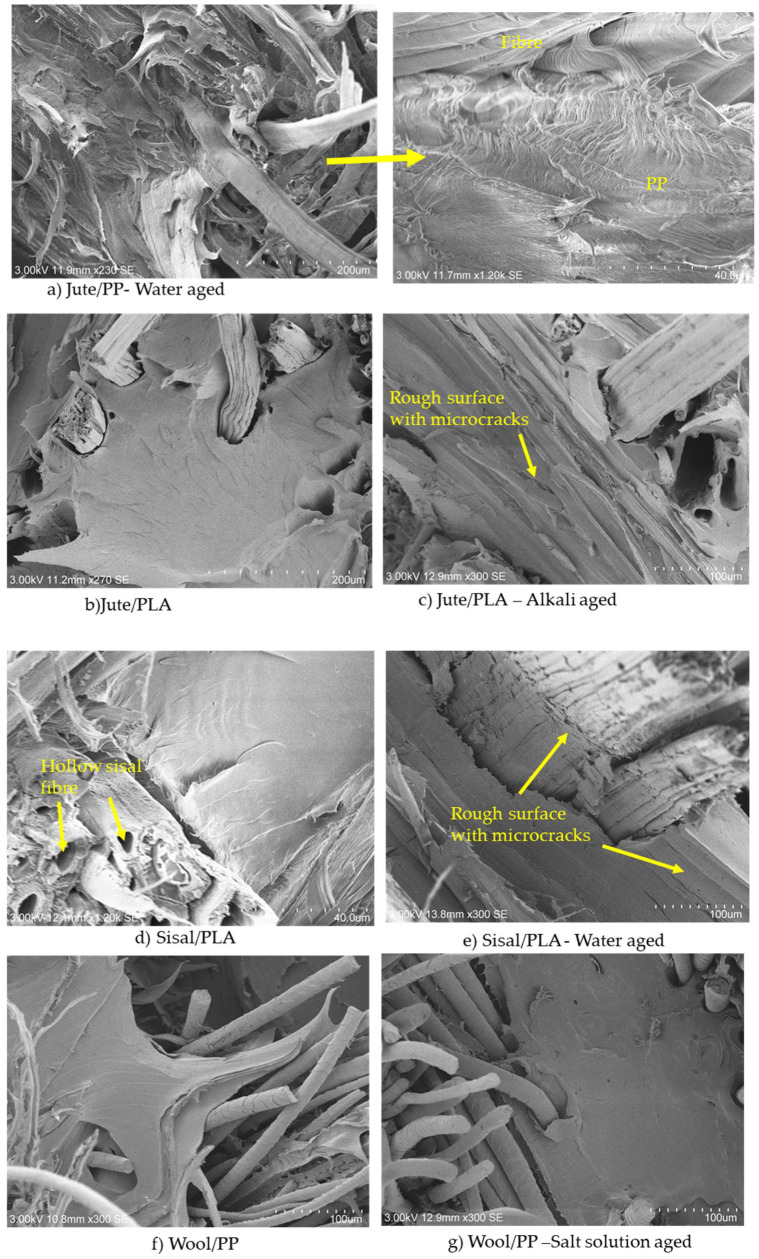
SEM images of the fractured surfaces of (**a**) jute/PP, (**b**) jute/PLA, (**c**) jute/PLA-alkali, (**d**) sisal/PLA, (**e**) sisal/PLA-water, (**f**) wool/PP and (**g**) wool/PP-salt solution exposure for 21 days.

**Figure 5 molecules-26-04581-f005:**
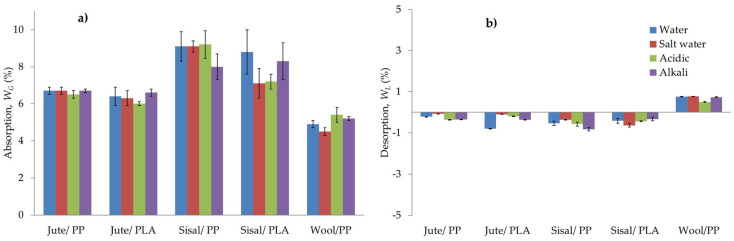
(**a**) Water/chemical solution absorption (*W_G_*) of composites after ageing in different solutions for 21 days and (**b**) desorption (*W_L_*) after drying at 50 °C for 24 h.

**Figure 6 molecules-26-04581-f006:**
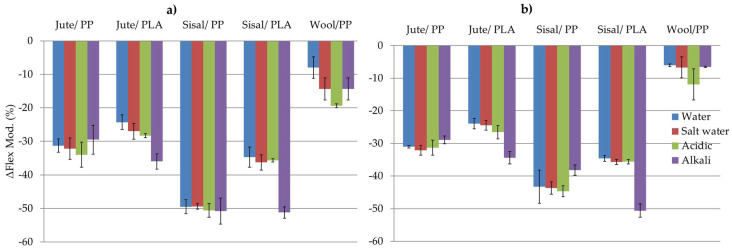
Change in flexural modulus between unaged and aged, ΔFlex: (**a**) air-dried and (**b**) oven-dried composite samples.

**Figure 7 molecules-26-04581-f007:**
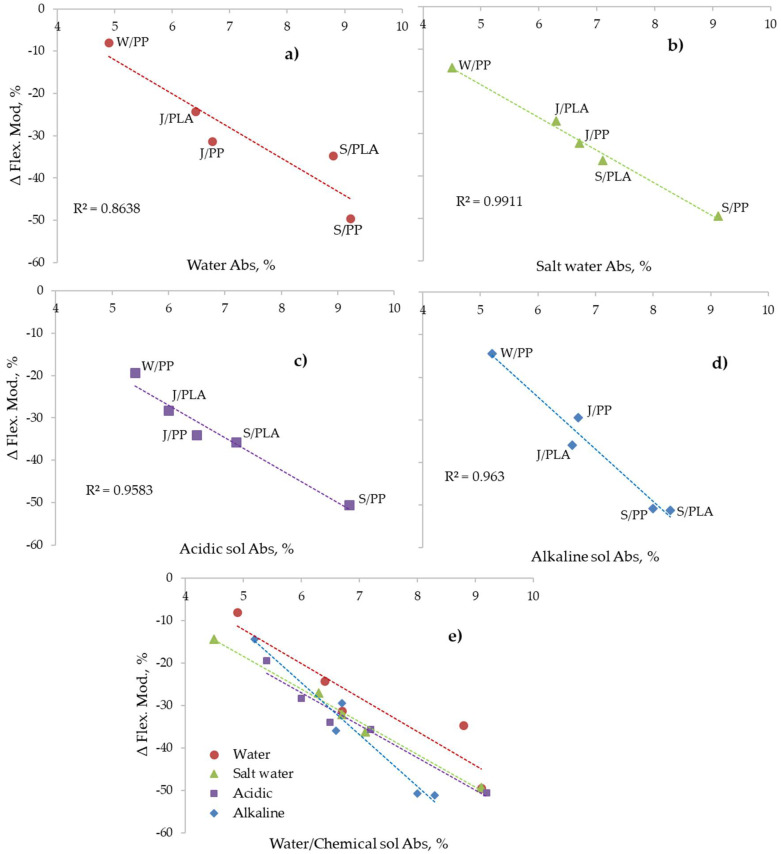
Relationship between (**a**) water, (**b**) salt water, (**c**) acidic, (**d**) alkali and (**e**) all water/conditions chemical solution absorption versus reduction in flexural modulus.

**Figure 8 molecules-26-04581-f008:**
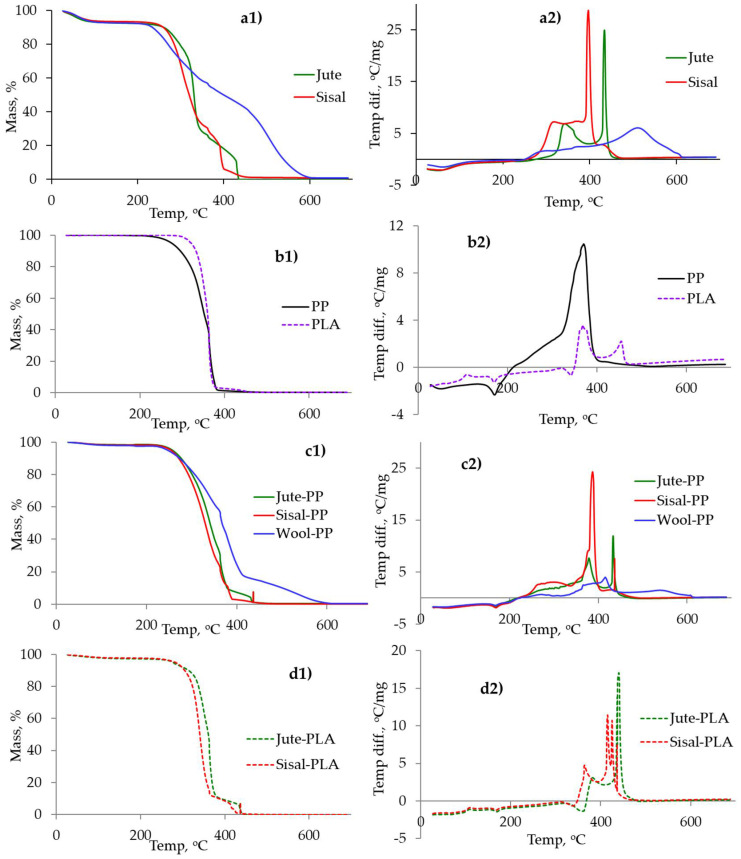
TGA (**a1**–**d1**)) and DTA (**a2**–**d2**) curves of (**a**) fibres, (**b**) polymeric matrices, (**c**) fibre/PP and (**d**) fibre/PLA composites.

**Figure 9 molecules-26-04581-f009:**
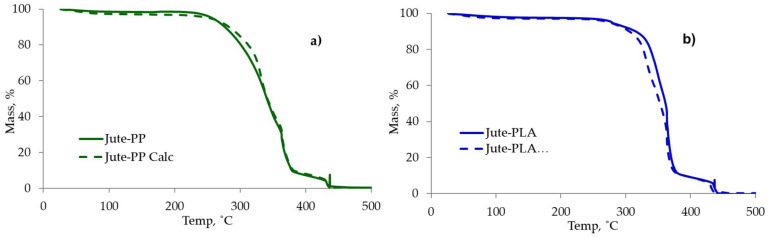
Experimental and calculated (dashed lines) thermogravimteric curves from sum of individual component curves of (**a**) jute/PP and (**b**) jute/PLA composites.

**Figure 10 molecules-26-04581-f010:**
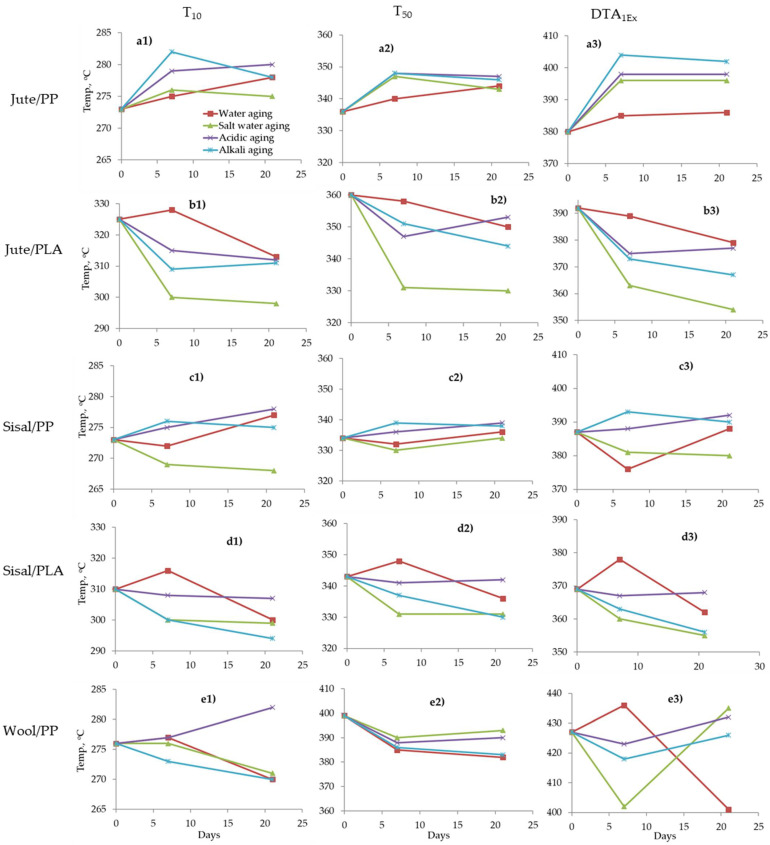
Effects of ageing on T_10_ (**a1**–**e1**), T_50_ (**a2**–**e2**) and DTA first exothermic peak maximum (**a3**–**e3**) of: (**a**) Jute/PP, (**b**) Jute/PLA, (**c**) sisal/PP, (**d**) sisal/PLA and (**e**) wool/PP composites.

**Table 1 molecules-26-04581-t001:** Composition of the composites.

Sample	Area Density of Reinforcing Fabric (g/m^2^)	Mass Fraction Fibre/Polymer (%)	Fibre Vol. Fraction (%)	Thickness (mm)
Jute/PP	174	42/58	31	3.2
Sisal/PP	62	41/59	30	1.3
Wool/PP	172	39/61	46	3.6
Jute/PLA	174	39/61	36	3.0
Sisal/PLA	62	34/66	31	1.3

**Table 2 molecules-26-04581-t002:** Water/chemical solution absorption properties of composites.

Composite	*W_G_*_(*Max*)_ (%)				D (mm^2^ s^−1^)			
	Water	Salt Water	Acidic	Alkaline	Water	Salt Water	Acidic	Alkaline
Jute/PP	6.3 ± 0.4	7.0 ± 0.2	7.2 ± 0.4	6.9 ± 0.1	4.3 × 10^−6^	7.9 × 10^−6^	5.7 × 10^−6^	5.1 × 10^−6^
Sisal/PP	8.0 ± 1.2	9.7 ± 0.8	6.2 ± 1.2	6.7 ± 1.0	1.3 × 10^−6^	2.1 × 10^−6^	7.8 × 10^−7^	1.0 × 10^−6^
Jute/PLA	6.0 ± 0.6	6.1 ± 0.6	5.4 ± 0.1	6.0 ± 0.1	4.6 × 10^−6^	1.5 × 10^−6^	1.7 × 10^−6^	4.6 × 10^−6^
Sisal/PLA	4.2 ± 2.4	5.7 ± 0.2	6.2 ± 0.8	12.2 ± 0.6	2.1 × 10^−6^	7.4 × 10^−7^	9.8 × 10^−7^	9.7 × 10^−7^
Wool/PP	6.5 ± 0.6	6.0 ± 0.6	6.0 ± 0.3	6.1 ± 0.2	1.1 × 10^−6^	2.5 × 10^−6^	1.2 × 10^−6^	2.5 × 10^−6^

**Table 3 molecules-26-04581-t003:** Mechanical and flammability properties of unaged and aged and oven-dried composites.

Composites	Ageing Conditions	Mechanical	Flammability				
		Flexural Modulus (GPa)	LOI (%)	Horizontal Burn			
				T_1_ (s)	T_2_ (s)	Rate (mm/s)	No. of Drips
Jute/PP	-	2.6 ± 0.4	18.7	115 ± 1	224 ± 3	0.44 ± 0.01	-
	Water	1.8 ± 0.2	18.9	122 ± 8	241 ± 11	0.42 ± 0.04	-
	Salt water	1.8 ± 0.1	19.3	115 ± 3	215 ± 5	0.45 ± 0.03	-
	Acidic	1.8 ± 0.1	18.9	103 ± 2	220 ± 1	0.47 ± 0.02	-
	Alkali	1.8 ± 0.3	19.1	96 ± 5	210 ± 15	0.50 ± 0.05	-
Jute/PLA	-	6.0 ± 0.4	19.8	80 ± 1	173 ± 1	0.62 ± 0.04	-
	Water	4.6 ± 0.2	19.9	84 ± 1	183 ± 2	0.57 ± 0.03	-
	Salt water	4.5 ± 0.6	20.1	80 ± 0	164 ± 5	0.62 ± 0.01	-
	Acidic	4.4 ± 0.2	19.9	87 ± 1	177 ± 1	0.57 ± 0.01	-
	Alkali	3.9 ± 0.1	19.9	78 ± 1	164 ± 1	0.63 ± 0.02	-
Sisal/PP	-	3.2 ± 0.3	18.7	64 ± 2	128 ± 1	0.78 ± 0.02	-
	Water	1.8 ± 0.1	18.8	63 ± 2	119 ± 2	0.82 ± 0.05	-
	Salt water	1.8 ± 0.1	19.2	58 ± 1	115 ± 1	0.87 ± 0.01	-
	Acidic	1.8 ± 0.2	18.8	60 ± 1	120 ± 1	0.84 ± 0.02	-
	Alkali	2.0 ± 0.1	19.0	57 ± 3	115 ± 1	0.88 ± 0.05	-
Sisal/PLA	-	5.9 ± 0.4	19.4	66 ± 3	126 ± 2	0.78 ± 0.02	-
	Water	3.9 ± 0.2	19.5	69 ± 3	121 ± 5	0.75 ± 0.09	-
	Salt water	3.8 ± 0.2	19.9	58 ± 1	115 ± 1	0.88 ± 0.01	-
	Acidic	3.8 ± 0.2	19.5	61 ± 1	119 ± 1	0.84 ± 0.02	-
	Alkali	2.9 ± 0.1	19.5	52 ± 2	107 ± 3	0.96 ± 0.03	-
Wool/PP	-	2.2 ± 0.3	20.7	144 ± 8	310 ± 15	0.34 ± 0.03	206 ± 15
	Water	2.1 ± 0.1	20.8	159 ± 3	390 ± 3	0.28 ± 0.02	216 ± 9
	Salt water	2.1 ± 0.2	20.8	150 ± 2	351 ± 5	0.31 ± 0.03	245 ± 12
	Acidic	1.9 ± 0.1	20.8	161 ± 8	400 ± 10	0.28 ± 0.08	196 ± 15
	Alkali	2.1 ± 0.1	20.8	160 ± 4	345 ± 6	0.30 ± 0.04	244 ± 10

**Table 4 molecules-26-04581-t004:** Thermal analysis data for fibres, polymer matrices and composites.

Sample	DTA Analysis		TGA Analysis			
	Melting Endotherm (°C)	Decomposition Peaks * (°C)	DTG Peak Maxima (°C)	T_10_ (°C)	T_50_ (°C)	Char at 500 °C (%)
PP	169	382, 423	381, 418	299	353	0.2
PLA	58, 169	342 (En), 376, 455	363, 453	332	360	0.4
Jute fibre	-	344, 433	285, 333, 438	262	332	0
Sisal fibre		316 (b), 396, 432(s)	328; 398, 437(d)	262	320	1.0
Wool fibre		295(b), 512	276, 521	237	401	26.1
Jute/PP	168	380, 439	352, 442	273	336	0.3
Jute/PLA	58, 169	360 (En), 392, 440	358, 453	325	360	0.2
Sisal/PP	167	387, 280(b), 387, 429	339, 427	273	334	0.2
Sisal/PLA	169	340(En), 369, 415	345, 432	310	343	0.4
Wool/PP	168	271(b), 426, 542	380,546	276	399	10.0

Note: * All peaks are exothermic in nature except for those denoted as En, endothermic. B = broad, ill defined peak. T_10_ = Temp. at 10% mass; T_50_ = Temp. at 50% mass.

## Data Availability

The data presented in this study are available on request from the corresponding author.

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
