# Peer review of "Effects of Water and Chemical Solutions Ageing on the Physical, Mechanical, Thermal and Flammability Properties of Natural Fibre-Reinforced Thermoplastic Composites"

_molecules, 2021, doi:10.3390/molecules26154581_

Round 1
Reviewer 1 Report
The article deals with the effect of jute, sisal and wool fiber as the reinforcement for fibre-reinforced plastic PP and PLA composites. These samples were processed by solvents and analyzed on water uptake, mechanical properties, and flammability point of view.
The article is interesting, well written, and it is characterized by a schematic and in-depth analysis from the difference in natural fibers and polymers point of view of the properties of these fiber-reinforced plastics, However, this high detail has not been maintained with regard to the experimental design and experiment analysis results. In particular, it is described that it is described that PP and PLA are analyzed respectively, there is insufficient information on the correlation between PP and PLA. In addition, it is difficult to detect significant results differences in Figures.
For this reason, in my opinion, the article is noteworthy of publication, but it needs major revisions listed below.
As both PP and PLA are covered in this article, the clarity of the experiment is not shown. In order to clearly show the results, the title, Figures need to be revised. (ex. relative mechanical properties analysis, correlation analysis, etc.) Among the many results, the most important results are shown in the manuscript, other results should be represented by Supplementary data.
It is difficult to find out differences in the photos of samples before and after solution absorbing especially sisal, and wool samples at Figure 2.
There is no evidence of the effect of voids on the color changing. It needs the supporting information or references about the forming of void.
The SEM images of Sisal/PLA of Figure 4 has different magnitude, and it is difficult to find out the surface morphology changes such as roughness and microcracks.
Graphs in Figure 3, and Figure 5 have no error bars. Furthermore, graphs in Figure 3 have improper solid curves which means exact values.
The reason why the WG of the Sisal/PLA sample are very different by the solvent is not fully explained. It needs references or an absorption mechanism.
The graphs of Figure 7 should be changed into another forms. Because of that there are no logical relationships of Flex. Mod/water Abs by the samples (W/PP, j/PLA, J/PP, S/PLA, S/PP). Therefore, dashed line cannot be used.
There are no significant changes of flammability properties by the fiber-reinforcement. Therefore, this result is not proper to prove the effects of this studies.
Reviewer 2 Report
This is an interesting manuscript with a topic falling within the study of natural fibre reinforced composites, that is, PP and PLA reinforced with jute, sisal and wool fibres. The effects of water, saltwater, acidic and alkali solutions ageing on water uptake, mechanical properties and flammability of the composites are reported. This is an interesting piece of work with some good results that could have some practical applications. In general, the results are clear, well presented and discussed, and the manuscript is well written. The following comments should be addressed before it can be recommended for publication:
Abstract
-There is a typo in Line 20 “upto” should be “up to”.
-I recommend adding one or two more lines in the abstract describing the results.
Introduction
-In the first paragraph, to put the research in context, the authors could add other examples of natural fibre reinforced composites for different applications. I recommend adding the following references:
https://doi.org/10.3390/ma13143060
https://doi.org/10.3390/fib6030053
-In the last (or penultimate) paragraph of the introduction, the authors should emphasize what is new and unique in this work that has not been done before. i.e. what is the novelty? Why is important?
General comments:
-In general, most of the graphs need improvement. The font size is too small in some figures, which makes them hard to read.
Section 2.1.2
-In Figure 6, the authors mention “Change in flexural modulus, ΔFlex, of unaged and aged…”; however, it seems that only the aged composites are shown in the graphs. Please double-check this.
Section 2.2.
-Table 3 should be placed on a single page, not across two different pages.
Section 4
-The authors could add some images to illustrate some of the methods and samples.
Round 2
Reviewer 1 Report
Thank you for your effort to revise the manuscript based on my comments, some of them were not fully corrected (Please see the attachment). Based on the review reports, I have made the decision to reject the manuscript in its current version. You still have the option to perform all the major changes and resubmit the manuscript to Molecules. I can find out your efforts to respond my notes, however, it is needed to describe it in the manuscripts more. Furthermore, you need to check your data and descriptions more precisely such as table and figure numbers in the response letter. Thank you.
